# Extra Virgin Olive Oil-Based Formulations: A “Green” Strategy against *Chlamydia trachomatis*

**DOI:** 10.3390/ijms241612701

**Published:** 2023-08-11

**Authors:** Marisa Di Pietro, Simone Filardo, Roberto Mattioli, Giuseppina Bozzuto, Agnese Molinari, Luciana Mosca, Rosa Sessa

**Affiliations:** 1Department of Public Health and Infectious Diseases, Section of Microbiology, Sapienza University of Rome, 00185 Rome, Italy; marisa.dipietro@uniroma1.it (M.D.P.); rosa.sessa@uniroma1.it (R.S.); 2Department of Biochemical Sciences, Faculty of Pharmacy and Medicine, Sapienza University of Rome, 00185 Rome, Italy; roberto.mattioli@uniroma1.it (R.M.); luciana.mosca@uniroma1.it (L.M.); 3National Centre for Drug Research and Evaluation, Italian National Institute of Health, 00161 Rome, Italy; giuseppina.bozzuto@iss.it (G.B.); agnese.molinari@iss.it (A.M.)

**Keywords:** OOP, oleacein, oleocanthal, NaDES, nutraceuticals, *Chlamydia trachomatis*, antimicrobial activity

## Abstract

In recent decades, antibiotic misuse has emerged as an important risk factor for the appearance of multi-drug-resistant bacteria, and, recently, antimicrobial resistance has also been described in *Chlamydia trachomatis* as the leading cause of bacterial sexually transmitted diseases worldwide. Herein, we investigated, for the first time, the antibacterial activity against *C. trachomatis* of a polyphenolic extract of extra virgin olive oil (EVOO), alongside purified oleocanthal and oleacein, two of its main components, in natural deep eutectic solvent (NaDES), a biocompatible solvent. The anti-chlamydial activity of olive-oil polyphenols (OOPs) was tested in the different phases of chlamydial developmental cycle by using an in vitro infection model. Transmission and scanning electron microscopy analysis were performed for investigating potential alterations of adhesion and invasion, as well as morphology, of chlamydial elementary bodies (EBs) to host cells. The main result of our study is the anti-bacterial activity of OOPs towards *C. trachomatis* EBs down to a total polyphenol concentration of 1.7 μg/mL, as shown by a statistically significant decrease (93.53%) of the total number of chlamydial-inclusion-forming units (*p* < 0.0001). Transmission and scanning electron microscopy analysis supported its anti-chlamydial effect, suggesting that OOP might damage the chlamydial outer layers, impairing their structural integrity and hindering EB capability to infect the host cell. In conclusion, OOPs may represent an interesting alternative therapeutic option toward *C. trachomatis*, although further studies are necessary for exploring its clinical applications.

## 1. Introduction

*Chlamydia trachomatis* is an obligate intracellular pathogen with a peculiar biphasic developmental cycle occurring into a phagocytic vacuole termed inclusion [1]. It alternates between two different forms, the extracellular infectious elementary body (EB), responsible for adhesion and invasion into the host cell, and the intracellular replicative reticulate body (RB), responsible for multiplication within the inclusion [2]. The entire duration of the *C. trachomatis* developmental cycle ranges approximately between 36 and 48 h, ending with the release of chlamydial EBs from infected cells and their dissemination within the host [3].

*C. trachomatis* is the leading cause of bacterial sexually transmitted diseases worldwide, with more than 130 million new cases per year according to the most recent World Health Organization (WHO) estimates [4]. The primary manifestations of *C. trachomatis* genital infection are urethritis, cervicitis, and salpingitis in women, although the majority of infections are asymptomatic [5,6]. Indeed, approximately 80% of women with *C. trachomatis* genital infection remain undiagnosed and untreated, potentially leading to several complications with serious consequences, including pelvic inflammatory disease, ectopic pregnancy, and obstructive infertility [7].

In recent years, the phenomenon of antibiotic resistance has also been described in *C. trachomatis*, and this might be related to the few classes of antibiotics recommended by the most recent guidelines, such as doxycycline, azithromycin, or levofloxacin [8]. Higher rates of treatment failures have been associated with treatment with azithromycin as compared to doxycycline towards *C. trachomatis* urogenital infections; drug-resistant *Chlamydia* serovars have been hypothesized as one of the reasons for azithromycin inefficiency, although another possible explanation may be represented by re-infection, non-adherence to therapeutic regimen, inadequate exposure to the antimicrobial drugs as well as the persistence of *C. trachomatis* infection [9]. However, the in vitro demonstration of diverse mutations in *C. trachomatis* associated with resistance to antibiotics supports the potential emergence of drug resistance in this pathogen [10].

Therefore, it is of great importance to explore novel therapeutic approaches toward *C. trachomatis*, and efforts are now focusing on the study of natural products as promising alternatives for the preparation of pharmacological formulations and nutraceuticals [11,12,13,14]. On this regard, the extra virgin olive oil (EVOO) obtained from cold pressure of *Olea europaea* L. drupes, a key component of the Mediterranean diet, is widely studied for its nutritional and therapeutic properties, especially for the prevention of cardiovascular diseases and metabolic disorders [15,16].

Traditionally, health-beneficial effects have been largely attributed to the high concentration of monounsaturated fatty acids (98–99% of the total weight of EVOO), although, in recent years, other components have been investigated, particularly olive-oil polyphenols (OOP) with promising antimicrobial properties [17,18,19,20]. Moreover, plant-based bioactive compounds have been previously reported to perform synergism with antibiotics in suppressing the growth of multi-drug-resistant bacteria by disrupting function and structure of the membrane phospholipid bilayer [21].

Different classes of polyphenols are present in EVOO, such as phenolic alcohols (e.g., tyrosol and hydroxytyrosol), phenolic acids (e.g., vanillic acid, ferulic acid and coumaric acid), flavonoids (e.g., apigenin and luteolin), lignans (e.g., pinoresinol) and, most importantly, secoiridoids, particularly oleocanthal and oleacein, which have been described to exert a significant antibacterial effect toward several bacterial species and *Candida albicans* [22].

In a previous paper, we demonstrated the inhibitory activity of OOP-based formulations in natural deep eutectic solvents (NaDES), which are biocompatible solvents, towards several drug-resistant Gram-positive and -negative strains, as well as *Candida* spp. as an interesting strategy that might help in reducing the risk of development of drug resistance [23].

Herein, we investigated, for the first time, the anti-bacterial activity of OOP extracted by NaDES, as well as that of purified oleocanthal and oleacein, dissolved in NaDES, toward *C. trachomatis*, an obligate intracellular bacterium. In particular, the anti-chlamydial effects of these compounds were assayed in the different phases of the chlamydial intracellular developmental cycle, and their adhesion and invasion to host cells, as well as changes in chlamydial EBs morphology, were also observed via scanning and transmission electron microscopy.

## 2. Results

### 2.1. OOP Composition

OOPs were obtained from EVOO of Coratina cultivar plants, a cultivar with a high quantity of polyphenols [24], by betaine/propylene glycol NaDES extraction. The chromatographic analysis by UHPLC-DAD/MS showed four main components identified as hydroxytyrosol, tyrosol, oleacein and oleocanthal, similar to previous preparations [23]. Quantitative analysis of the chromatographic peaks showed that oleacein and oleocanthal are present in large amounts compared to the other polyphenols. In particular, the concentrations of hydroxytyrosol, tyrosol, oleacein, and oleocanthal in OOP were 0.39 ± 0.06 mg/mL, 0.99 ± 0.17 mg/mL, 4.35 ± 0.21 mg/mL, and 9.61 ± 0.62 mg/mL, respectively (Appendix A). Furthermore, the total polyphenol content, measured using the Folin–Ciocalteu assay and expressed as hydroxytyrosol equivalents, is 16.95 ± 2.22 mg/mL. The total polyphenol content and the single-compound concentrations of OOP are reported in Table 1.

### 2.2. Cytotoxicity of OOP, Oleacein and Oleocanthal

Preliminary investigations were carried out to determine the effect of OOP, oleacein and oleocanthal, dissolved in NaDES, on the viability of McCoy cells, in order to verify that these substances did not exert a direct cytotoxic effect on cells and to optimize the amount of mixture to be used in anti-chlamydial experiments.

As shown in Figure 1, a statistically significant decrease in McCoy cell viability was evidenced for OOP, oleacein and oleocanthal in NaDES starting from a 1:200 dilution ratio (corresponding to a polyphenol concentration of 85 μg/mL for OOP, and a concentration of 0.25 mM for oleacein and oleocanthal). Interestingly, oleacein showed a lower cytotoxicity on McCoy cells as compared to oleocanthal at the concentrations of 0.5 mM (32.03% decrease in cell viability vs. 84.00%, respectively, *p* < 0.0000001) and 0.25 mM (16.51% decrease in cell viability vs. 71.78%, respectively, *p* < 0.00001).

NaDES alone also showed a statistically significant cytotoxicity on McCoy cells up to the dilution ratio 1:200 (15.6% decrease in cell viability), albeit with a lower reduction of cell viability than that observed in the presence of OOP (58%), oleacein (16.5%) and oleocanthal (71.8%), as shown in Appendix A.

### 2.3. Anti-C. trachomatis Activities of OOP, Oleocanthal and Oleacein

Initially, the anti-bacterial properties of NaDES alone against *C. trachomatis* were investigated in the same experimental conditions as the EVOO-based formulations; no statistically significant decrease in the total number of chlamydial IFUs was observed at either the pre-treatment, pre-incubation or treatment phases (Appendix A).

Then, the anti-chlamydial activity of OOP, oleacein and oleocanthal was evaluated directly against extracellular *C. trachomatis* EBs (pre-treatment phase), at the concentrations reported in Table 2. A significant reduction of chlamydial infectious EBs was observed for OOP at all the concentrations investigated (*p* < 0.05, Figure 2 and Table 2), as well as for oleocanthal and oleacein in NaDES (*p* < 0.01 and *p* < 0.05, respectively, Figure 3 and Table 2), as compared to untreated *C. trachomatis* EBs. Furthermore, the anti-chlamydial activity showed a dose-dependent effect, as evidenced by a decrease in chlamydial inhibition at lower concentrations (Table 2).

Lastly, we evaluated the anti-chlamydial activity of OOP, oleacein and oleocanthal in NaDES by treating McCoy cells before (pre-incubation phase) and after (treatment phase) *C. trachomatis* infection at non-cytotoxic concentrations: 17 μg/mL and 1.7 μg/mL for OOP, and 0.05 mM and 0.005 mM for both oleacein and oleocanthal. As shown in Figure 4, no statistically significant decrease in the total number of *C. trachomatis* IFUs was observed in the pre-incubation and treatment phases for OOP, oleacein and oleocanthal in NaDES at all the concentrations tested.

### 2.4. Transmission Electron Microscopy of C. trachomatis EB Adhesion and Invasion

Given the high anti-chlamydial activity observed when *C. trachomatis* EBs were pre-treated with OOP in NaDES, at a concentration of 170 μg/mL, we aimed to investigate whether this effect could alter chlamydial adhesion and invasion of host-cell membranes and, hence, the development of nascent inclusions via transmission electron microscopy.

As shown in Figure 5, untreated *C. trachomatis* EBs are able to adhere to the cell membrane, followed by their internalization via endocytosis, as evidenced by the presence of chlamydial EBs into the host-cell cytoplasm. On the contrary, the treatment of *C. trachomatis* EBs with OOP renders them unable to adhere to and invade the host cell, as demonstrated by the absence of chlamydial EBs at the cell surface or within the host cytoplasm.

### 2.5. Scanning Electron Microscopy of OOP-Treated C. trachomatis EBs

Given the absence of chlamydial EB adhesion and invasion within host cells following pre-treatment with OOP at the concentration of 170 μg/mL, as evidenced by transmission electron microscopy, we aimed to investigate their potential ultrastructural changes via scanning electron microscopy.

As shown in Figure 6, untreated *C. trachomatis* EBs appear as clusters of spherical particles with a diameter of approximately 200 to 300 nm, consistent with the typical *C. trachomatis* EB size as reported in the literature [25]. Conversely, scanning electron micrographs of *C. trachomatis* EBs treated with 170 μg/mL of OOP show the presence of a network of aberrant spherical particles with a much smaller diameter (5–10 nm), indicating the absence of morphologically normal chlamydial EBs.

## 3. Discussion

Herein, for the first time, the anti-microbial activity of different green OOP-based formulations in NaDES towards *C. trachomatis* has been investigated. The main result of our study is compelling evidence that OOP possessed a potent anti-chlamydial activity down to a total non-cytotoxic polyphenol concentration of 1.7 μg/mL, with a dose-dependent effect. Similarly, its purified active compounds, oleacein and oleocanthal, also showed a significant anti-chlamydial activity. In particular, OOP-based formulations were effective towards *C. trachomatis* EBs, the extracellular forms responsible for transmission of the infection and dissemination into the host, limiting, hence, the development of chronic complications. Interestingly, OOP was characterized by the highest anti-chlamydial activity, as evidenced by a >90% reduction of *C. trachomatis* IFUs. The anti-microbial effect against chlamydial EBs was confirmed by transmission electron microscopy analysis, revealing the absence of chlamydial EBs attached to cell membranes or internalized in phagocytic vacuoles after OOP treatment. Scanning electron microscopy also supported anti-chlamydial EB activity, highlighting chlamydial extracellular forms with a consistent decrease in their size following treatment with OOP, suggesting that it might damage the chlamydial outer layers, impairing their structural integrity.

The discovery of novel “green” OOP-based formulations as anti-chlamydial agents is highly relevant in light of the fact that, in recent years, the problem of antimicrobial resistance has also acquired growing importance in *C. trachomatis*, as suggested by significant treatment failure rates following first-line antimicrobials like doxycycline or azithromycin [26]. In particular, treatment failure ranged from 5% to 23% depending upon the population examined, and further evidence showed higher treatment failure rates (up to 14%) during azithromycin treatment than with doxycycline [9,27,28]. These observations are not surprising since in vitro studies have reported that azithromycin resistance could be raised through mutations in *C. trachomatis* 23S rRNA genes [10,29]. Moreover, other genetic mutations responsible for resistance to different microbial drugs were described, including, for example, tet(M) gene for tetracycline resistance [10,28,30]. Consequently, the phenomenon of *C. trachomatis* antibiotic resistance is highly likely, although, to date, no chlamydial resistant strains have been isolated; this may be explained, indeed, by the relatively low sensitivity of culture methods for the isolation of *C. trachomatis* (up to 50%), especially when compared to modern nucleic acid amplification tests [31,32].

More importantly, there are increasing concerns that tetracycline resistance, found in *C. suis* isolates from pigs and then associated with the resistance gene tetC [33,34], can be transferred into *C. trachomatis* strains. Indeed, gene transfer between *C. suis* and *C. trachomatis* can occur in nature, as evidenced by the observation of 16s rRNA genes from *C. suis*, in different *C. trachomatis* isolates, and by in vitro studies showing horizontal transfer of tetR from *C. suis* to human clinical isolates of *C. trachomatis* following co-culture [35,36].

Interestingly, in our study, OOP showed higher anti-chlamydial activity as compared to its main active components alone, oleacein and oleocanthal, and this may be attributed to the extraction method using NaDES, a mix of natural compounds (betaine:propylene glycol), that has led to an increased yield of secoiridoids. As a matter of fact, the main advantage of using polyphenols from EVOO in NaDES compared to single purified polyphenols is the lower production cost of the extract compared to individual compounds, making it accessible for in vivo experiments or human use. Furthermore, the extraction of polyphenols from EVOO using NaDES presents significant advantages over the use of traditional organic solvents typically employed for the preparation of these extracts. NaDES are biocompatible “green” solvents that pose no risks to humans or the environment, unlike, for example, organic solvents such as methanol. Moreover, the preparation time of the extracts is considerably reduced, since in a single step, it is possible to obtain a concentrated extract of polyphenols without the need to remove the organic solvent [37]. This is, indeed, an efficient preparative method that allows one to obtain concentrated extracts with an easy, rapid, and eco-friendly procedure, that can be applied to many other plants, food matrices or food wastes for the extraction of bioactive molecules.

Limitations of this study include the lack of synergism tests with the antibiotics recommended for treating chlamydial genital infections, like azithromycin or doxycycline, as well as drug-resistant strains of *C. trachomatis*.

In conclusion, the promising anti-chlamydial properties of OOP, alongside the biocompatible and eco-friendly extraction method with a low impact on human health and the environment, suggest this formulation as a potential alternative therapeutic option for *C. trachomatis* genital infections, although further studies are necessary for exploring its clinical applications.

## 4. Materials and Methods

### 4.1. Preparation and Analysis of the Antimicrobial Agents

Polyphenolic extract was prepared from extra virgin olive oil (EVOO) obtained by cold pressing of the drupes of *Olea europaea* L. (Coratina cultivar), purchased from a local market in the Puglia region (Corato, Italy). The polyphenolic fraction was extracted as described in [37]. The NaDES was prepared by mixing two components (betaine:propylene glycol 1:3.5 molar ratio) at 70 °C under magnetic stirring for 1 h. After cooling, EVOO was added to the NaDESs in a 1:50 *v*/*v* (NaDES:EVOO) ratio. The extraction was carried out under magnetic stirring at room temperature for 15 min and then transferred via a separatory funnel for decantation and phase separation. The extract was analysed to determine the total polyphenol content and polyphenolic composition. Total phenols were determined by the Folin–Ciocalteu assay [38] by mixing 790 μL of distilled water with 10 μL of standard, sample, or blank. To these, 50 μL of Folin–Ciocalteu reagent (Merck KGaA, Darmstadt, Germany) was added, incubated for 3 min at room temperature, and then 150 μL of 20% (*w*/*v*) Na_2_CO_3_ was added. After 2 h of incubation, the absorbance at 760 nm was measured using a Hitachi U2000 spectrophotometer (Hitachi, Tokyo, Japan). The results were expressed as hydroxytyrosol equivalents. The extracts were also assayed by UHPLC-DAD/MS, by utilizing a Waters Acquity H-Class UPLC system, as previously described by Francioso et al. (2020) [37]. The chromatographic system was coupled to a photodiode array (DAD PDA) and a single-quadruple mass detector with an electrospray ionization source (QDa). Chromatography was performed on a reverse-phase C18 column (Phenomenex Kinetex, 100 mm × 2.1 mm i.d., 2.6 μm particle size). Solvent A was 0.1% aqueous formic acid (Merck), and solvent B was 0.1% formic acid in methanol (UPLC gradient grade, Merck). The flow rate was 0.5 mL/min, and the column temperature was set at 35 °C. Elution was performed with a linear gradient from 2 to 100% B in a total time of 17 min including re-equilibration. The samples were diluted in the mobile phase and injected through the needle. The photodiode array detector was set up in the range of 200–600 nm. Mass spectrometric detection was performed in the negative electrospray ionization mode, using nitrogen as the nebulizer gas. Analyses were performed in the total ion current (TIC) mode with a mass range of 50–1000 *m*/*z*. The capillary voltage was 0.8 kV, cone voltage 15 V, ion source temperature 120 °C, and probe temperature 600 °C. Compounds were identified by retention time, *m*/*z*, UV-VIS spectrum, and comparing them with commercially available standards (Merck). Quantification of each compound was performed by using standard calibration curves in the range of 0.1–2 nmol. Oleacein and oleocanthal single molecules, were provided by Active-Italia S.r.l. freeze-dried and dissolved in NaDES at final concentration of 50 mM (16.0 mg/mL and 15.2 mg/mL for oleacein and oleocanthal, respectively).

### 4.2. Cell Culture and Culture Conditions

The McCoy cell line (ECACC, Public Health England, catalogue number 90010305, Porton Down, Salisbury, UK) was seeded in 25 cm^2^ cell culture flasks with ventilated caps and grown in Dulbecco’s Modified Eagle Medium (DMEM) supplemented with 10% foetal bovine serum (FBS) at 37 °C in a humidified atmosphere with 5% CO_2_. Upon confluency (>85%), cells were passaged with brief trypsinization.

### 4.3. Propagation and Titration of C. trachomatis

*C. trachomatis* serovar D strain UW3 (VR-855, ATCC, Manassa, VA, USA) was propagated in McCoy cells, as previously described [1]. Briefly, confluent McCoy cell monolayers were infected with chlamydial EBs by centrifugation at 750× *g* for 30 min, and then harvested by scraping after 36 to 40 h post infection. The resulting suspension was, then, vortexed with sterile glass beads for 2 min and, after the removal of cell debris, the supernatant, containing chlamydial EBs, was added to an equal volume of 4× Sucrose Phosphate (4SP) buffer, and stored at −80 °C.

*C. trachomatis* titration was performed via immunofluorescence assay (IFA); briefly, McCoy cell monolayers were infected with 10-fold serial dilutions of bacterial stock, incubated for 48 h at 37 °C in humidified atmosphere with 5% CO_2_, fixed with methanol and stained with isothiocyanate-conjugated monoclonal antibody anti-*C. trachomatis* LPS (Merifluor^®^ *Chlamydia*, Meridian Bioscience Inc., Cincinnati, OH, USA), as previously described [39]. The total number of *C. trachomatis* inclusion-forming units (IFUs) was enumerated by counting all microscope fields using a fluorescence microscope (400× magnification).

### 4.4. Cytotoxicity of OOP

Confluent McCoy cell monolayers, grown on 96-well cell culture trays, were incubated with increasing concentrations of OOP, oleacein and oleocanthal in NaDES (dilution ratios of 1:25, 1:50, 1:100, 1:200, 1:1000 and 1:10,000 (*v*/*v*) in DMEM supplemented with 10% FBS, corresponding to a total polyphenol concentration range of 680 μg/mL to 1.7 μg/mL for OOP, and 2 mM to 0.005 mM for oleacein and oleocanthal) at 37 °C in humidified atmosphere with 5% CO_2_. Cytotoxicity of NaDES alone was also investigated at the same dilution ratio and incubation conditions. After 24 h, the number of viable cells was determined via MTT (3-(4,5-dimethylthiazol-2-yl)-2,5-diphenyltetrazolium bromide, a tetrazole) assay, as previously described [11].

### 4.5. Effects of OOP, Oleocanthal and Oleacein on the Different Phases of C. trachomatis Developmental Cycle

OOP, oleocanthal and oleacein in NaDES, as well as NaDES alone, were tested at increasing concentrations for their anti-chlamydial activity on the different phases of the *C. trachomatis* developmental cycle, specifically the pre-treatment of chlamydial EBs, the pre-incubation of McCoy cell monolayers followed by *C. trachomatis* EB infection, and the treatment of McCoy cell monolayers infected with *C. trachomatis* EBs.

#### 4.5.1. Pre-Treatment Phase

To detect the activity of OOP, oleocanthal and oleacein in NaDES, as well as NaDES alone, on *C. trachomatis* serovar D, 5000 EBs/mL, corresponding to a multiplicity of infection (MOI) of 0.05, were pre-incubated in DMEM supplemented with 10% FBS, in the absence or presence of OOP, oleocanthal and oleacein in NaDES for 1 h at 37 °C in humidified atmosphere with 5% CO_2_. Subsequently, *C. trachomatis* EB suspension, containing the tested compounds, was further diluted 10 times in DMEM supplemented with 10% FBS, and, hence, used to infect McCoy cell monolayers grown on glass coverslips in 24-well cell-culture trays by centrifugation at 750× *g* for 30 min at room temperature. Subsequently, the cells were washed with DPBS to remove non-internalized *C. trachomatis* EBs and newly incubated in fresh culture medium, consisting of DMEM supplemented with 10% FBS. After 36 h of incubation at 37 °C in a humidified atmosphere with 5% CO_2_, the total number of *C. trachomatis* IFU was determined by IFA.

#### 4.5.2. Pre-Incubation Phase

McCoy cell monolayers, grown on glass coverslips in 24-well cell-culture trays, were pre-incubated in DMEM supplemented with 10% FBS, in the absence or presence of OOP, oleocanthal and oleacein in NaDES, as well NaDES alone. After 24 h of incubation at 37 °C in a humidified atmosphere with 5% CO_2_, the cell culture medium containing the tested compounds was removed by washing the cells 3 times with DPBS, and then, McCoy cell monolayers were infected with *C. trachomatis* at a MOI of 0.05 as described above. After 36 h of incubation at 37 °C in humidified atmosphere with 5% CO_2_, the total number of *C. trachomatis* IFU was determined by IFA.

#### 4.5.3. Treatment Phase

McCoy cell monolayers, grown on glass coverslips in 24-well cell-culture trays, were infected with *C. trachomatis* at an MOI of 0.05, by centrifugation at 750× *g* for 30 min at room temperature. Subsequently, the cells were washed with DPBS to remove non-internalized *C. trachomatis* EBs and fresh medium, with or without OOP, oleocanthal and oleacein in NaDES, as well as NaDES alone, was added to the infected cells. After 36 h of incubation at 37 °C in humidified atmosphere with 5% CO_2_, the total number of *C. trachomatis* IFU was determined by IFA.

### 4.6. Scanning Electron Microscopy of Pre-Treated C. trachomatis EBs

*C. trachomatis* EBs at a MOI of 0.05 were pre-incubated in DMEM supplemented with 10% FBS, in the absence or presence of OOP in NaDES at the highest effective concentration, for 1 h at 37 °C in a humidified atmosphere with 5% CO_2_. Subsequently, 100 µL aliquots of each solution containing *C. trachomatis* EBs were placed on a polylysine-covered glass coverslip (12 mm diameter) and put to dry at room temperature for 3 h. Samples were then fixed with 2.5% glutaraldehyde in 0.1 M cacodylate buffer (pH 7.4) at room temperature for 1.5 h, post-fixed with 1% OsO_4_ in the same buffer for 2 h, dehydrated through a graded ethanol series, critical point dried with CO_2_ (CPD 030 Balzers device, Bal-Tec, Balzers, Pfäffikon, Switzerland), and gold coated by sputtering (SCD040 Balzers device, Bal-Tec). Samples were examined with a field emission gun scanning electron microscope (SEM-FEG, Quanta 200 Inspect, FEI Company, Eindhoven, The Netherlands).

### 4.7. Transmission Electron Microscopy of Pre-Treated C. trachomatis EBs

*C. trachomatis* EBs at an MOI of 0.05 were pre-incubated in DMEM supplemented with 10% FBS, in the absence or presence of OOP in NaDES at the highest effective concentration, for 1 h at 37 °C in humidified atmosphere with 5% CO_2_. Subsequently, *C. trachomatis* EB suspensions were used to infect McCoy cell monolayers, grown on glass coverslip in 24-well cell-culture trays, by centrifugation at 750× *g* for 30 min at room temperature. The cells were, then, washed with DPBS to remove non-internalized *C. trachomatis* EBs and newly incubated in DMEM supplemented with 10% FBS. After 2 h of incubation at 37 °C in humidified atmosphere with 5% CO_2_, cells were fixed in 2% glutaraldehyde and 0.5% paraformaldehyde in 0.1 M sodium cacodylate buffer containing 3 mM CaCl_2_ and 0.1 M sucrose (pH 7.4), at room temperature for 30 min, and stored at 4 °C. After waiting overnight, the fixed cells were scraped, washed in 0.15 M sodium cacodylate buffer containing 3 mM CaCl_2_ (pH 7.4) and centrifuged. The pellets were resuspended and postfixed in 2% osmium tetroxide in 0.07 M sodium cacodylate buffer containing 1.5 mM CaCl_2_ (pH 7.4) at 4 °C for 2 h, dehydrated through graded ethanol concentrations and embedded in Epon 812 resin (Electron Microscopy Science, Fort Washington, PA, USA). Ultrathin sections, obtained with a Leica UC6 ultramicrotome (Leica Microsystems, Wetzlar, Germany), were contrasted with UranyLess EM Stain (Electron Microscopy Sciences, Hatfield, PA, USA) and lead citrate, and examined with a Philips 208S transmission electron microscope (FEI Company, Eindhoven, The Netherlands).

### 4.8. Statistical Analysis

All values are expressed as means ± standard deviation (SD) of two to four replicates from at least two independent in vitro experiments. Comparisons of means were performed using a two-tailed Student t-test for independent samples. The single or multiple inference significance level was set to 5%. All statistical calculations and graphs were performed in the software Excel (Microsoft, Redmond, WA, USA, version 2302, build 16130.20332).

## Figures and Tables

**Figure 1 ijms-24-12701-f001:**
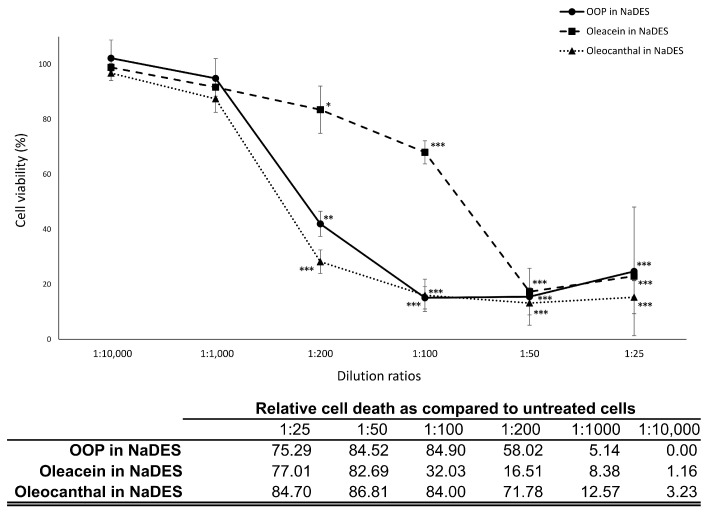
Cytotoxicity of OOP, oleacein and oleocanthal in NaDES. Cytotoxicity was evaluated at dilution ratios ranging from 1:25 to 1:10,000, corresponding to a range of concentrations from 680 μg/mL to 1.7 μg/mL for OOP, and from 2 mM to 0.005 mM for oleacein and oleocanthal. *, *p* < 0.05; **, *p* < 0.01; and ***, *p* < 0.001 vs. untreated cells.

**Figure 2 ijms-24-12701-f002:**
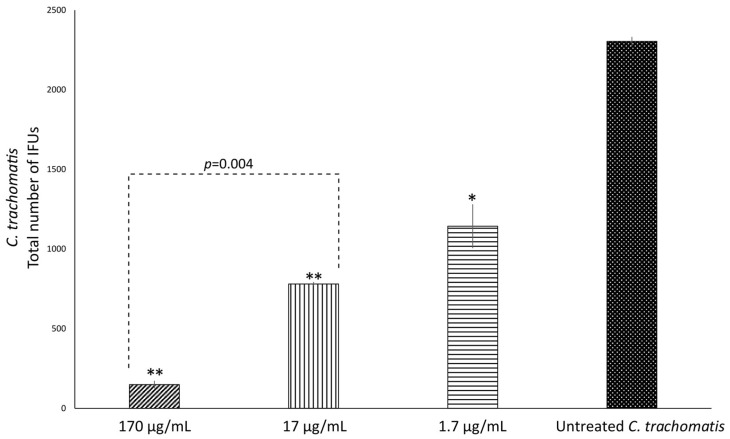
Anti-bacterial activity of OOP against *C. trachomatis* EBs. *C. trachomatis* EBs, at a MOI of 0.05, were first pre-treated in the presence or absence of different concentrations of OOP in NaDES for 1 h. McCoy cells were, then, infected with *C. trachomatis* EB suspensions and the total number of IFUs enumerated via fluorescence microscopy after 36 h of incubation. *, *p* < 0.05; **, *p* < 0.0001 vs. untreated *C. trachomatis* EBs.

**Figure 3 ijms-24-12701-f003:**
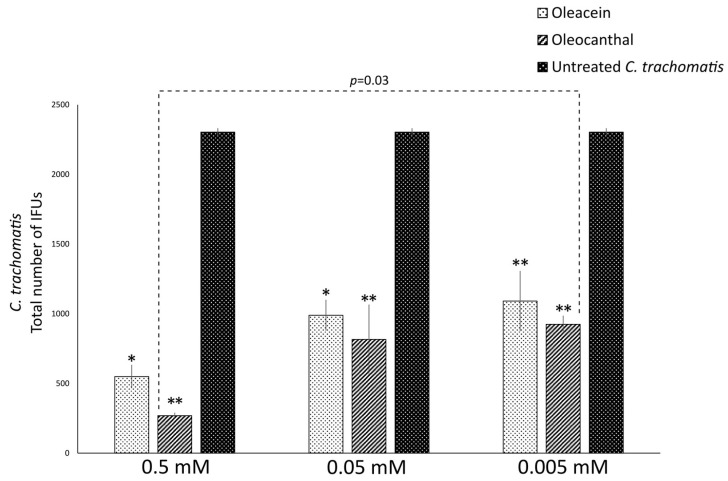
Anti-bacterial activity of oleacein and oleocanthal against *C. trachomatis* EBs. *C. trachomatis* EBs, at a MOI of 0.05, were first pre-treated in the presence or absence of different concentrations of oleacein or oleocanthal in NaDES for 1 h. McCoy cells were, then, infected with *C. trachomatis* EBs suspensions and the total number of IFUs enumerated via fluorescence microscopy after 36 h of incubation. *, *p* < 0.05; **, *p* < 0.01 vs. untreated *C. trachomatis* EBs.

**Figure 4 ijms-24-12701-f004:**
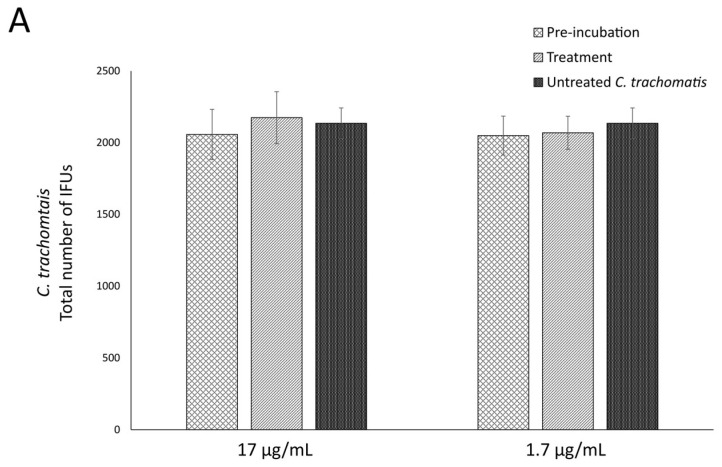
Anti-chlamydial activity of OOP, oleacein and oleocanthal at the pre-incubation and treatment phases. In the pre-incubation phase, McCoy cell monolayers were first exposed to different concentrations of OOP (**A**), oleacein (**B**) and oleocanthal (**C**) for 24 h, followed by *C. trachomatis* infection at a MOI of 0.05; in the treatment phase, McCoy cell monolayers were first infected with *C. trachomatis* at a MOI of 0.05 and then treated with different concentrations of all the compound tested. Total number of IFUs was enumerated following approximately 36 h of incubation via fluorescence microscopy.

**Figure 5 ijms-24-12701-f005:**
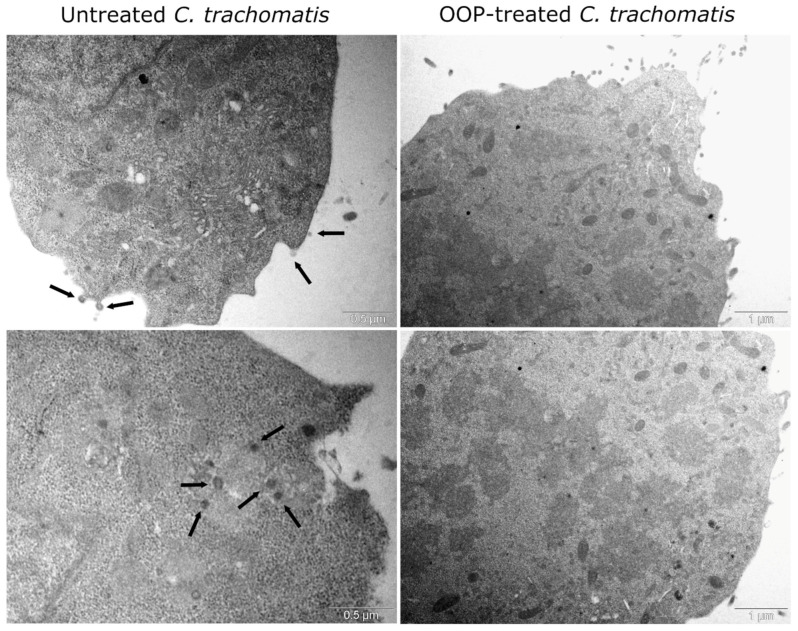
Transmission electron micrographs of *C. trachomatis* EBs treated with OOP. *C. trachomatis* EBs were pre-treated with OOP in NaDES (170 μg/mL) for 1 h, followed by infection of McCoy cells for 2 h. Then, infected cells were examined via transmission electron microscopy, at different magnification levels (1 μm and 0.5 μm). Micrographs show the adhesion of untreated *C. trachomatis* EBs to cell membrane and their invasion into the host-cell cytoplasm (arrows). Conversely, OOP-treated *C. trachomatis* EBs are unable to adhere to the cell membrane. Upper panels show the adhesion of chlamydial EBs to cell membrane, while the lower panels show the invasion of chlamydial EBs into host cell.

**Figure 6 ijms-24-12701-f006:**
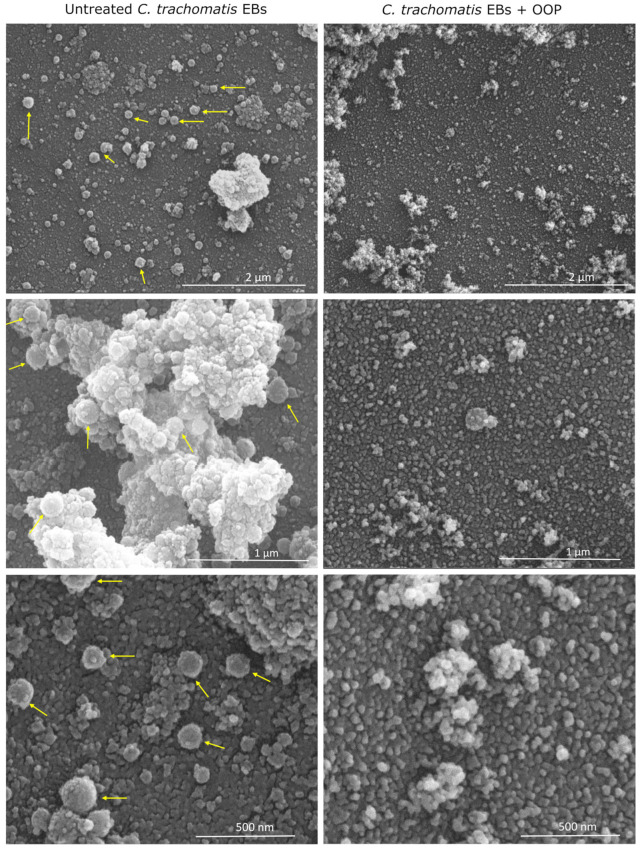
Scanning electron micrographs of *C. trachomatis* EBs treated with OOP. *C. trachomatis* EBs were pre-treated with OOP in NaDES (170 μg/mL). Then, chlamydial EBs were examined via scanning electron microscopy, at different magnification levels (2 μm, 1 μm and 0.5 μm). Yellow arrows point to the spherical particles whose size is compatible with that of morphologically normal *C. trachomatis* EBs.

**Table 1 ijms-24-12701-t001:** Total polyphenol content and single-compound concentrations of OOP in NaDES.

	Total Polyphenols	Hydroxytyrosol	Tyrosol	Oleacein	Oleocanthal
	(mg/mL)				
OOP in NaDES	16.95 ± 2.22	0.39 ± 0.06	0.99 ± 0.17	4.35 ± 0.21	9.61 ± 0.62

The values are reported as mean ± standard deviation.

**Table 2 ijms-24-12701-t002:** Anti-chlamydial activity of OOP, oleocanthal and oleacein against *C. trachomatis* EBs.

		Percentage of Inhibition		
		1:100	1:1000	1:10,000
OOP in NaDES		93.53	66.14	50.41
Oleacein in NaDES		76.15	57.05	53.45
Oleocanthal in NaDES		88.34	63.71	59.89

OOP: 170 μg/mL to 1.7 μg/mL; oleacein and oleocanthal: 0.5 mM to 0.005 mM.

## Data Availability

The data presented in this study are available in the article or Appendix A.

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
