# Peer review of "Extra Virgin Olive Oil-Based Formulations: A “Green” Strategy against *Chlamydia trachomatis"

_ijms, 2023, doi:10.3390/ijms241612701_

Round 1
Reviewer 1 Report
Review report
Manuscript ID: ijms-2544126
Title: Extra Virgin Olive Oil-based formulations: a “green” strategy against Chlamydia trachomatis
Authors: Marisa Di Pietro, Simone Filardo *, Roberto Mattioli, Giuseppina, Bozzuto, Agnese Molinari, Luciana Mosca, Rosa Sessa
Submitted to section: Molecular Microbiology, Antibacterial Activity of Drug-Resistant Strains.
In this work, the authors investigated the putative antichlamydial activity of Extra Virgin Olive Oil extract and compared it to the antibacterial activity of purified oleocanthal and oleacein, two of the main components of the Olive Oil extract. Studies were performed in a mixture of betaine and propylene glycol, a Natural Deep Eutectic Solvent, supposed to be biocompatible.
This work is particularly part of the fight against the resistance induced by certain drugs prescribed during infection by C. trachomatis.
Several elements challenged me.
11) In the introduction, the authors suggest that treatment failure, which can reach 23% of people treated, is due to a phenomenon of resistance. To announce this is to conceal the other reasons for this failure which are essential (re-infection, non-adherence to the treatment regimen, inadequate exposure to the antimicrobial and persistence phenomena).
2) The authors announce the use of a "green" solvent while its toxicity is demonstrated in figure S1. The choice of this solvent is then debatable. The choice of the betaine/propylene glycol mixture is not explained in the manuscript, I wonder about the possibility of replacing this mixture of solvents with a truly non-toxic solvent.
3) Did the authors wonder about the anti-chlamydial activity of the solvent ?
4) Line 86-87: the sentence seems incomplete. The word “extraction” is probably missing.
5) The authors announce a certain composition of this olive oil extract (hydroxytyrosol, tyrosol, oleacein and oleocanthal) but no experimental evidence is provided to attest to this composition. Authors must provide, supporting information, the chromatograms obtained and processed during the HPLC performed as well as the mass spectroscopy results.
6) Figure 1 provides the cytotoxicity results of the olive oil extract and its 2 main components. These are diluted in the solvent to ratios where the toxicity of the latter is identified (>1:100). Under these conditions, it is therefore not possible to discriminate between the toxicity induced by the compound and that of the solvent.
7) Table 2 provides the antibacterial activities of the compounds studied. The 1:100 column shows activity while Table 1 shows cellular toxicity of the compounds at this concentration. This activity is therefore linked to the cell death induced by the compounds and not to their antibacterial activity.
8) The authors announce that they want to fight against the phenomenon of resistance observed in C. trachomatis. However, no experimental evidence shows that the olive oil extracts used are active against resistant strains of the bacteria. Announcing, line 211, that isolating this type of strain is time-consuming is not a plausible argument in Science.
9) Line 220-222 : “Interestingly, in our study, OOP showed higher anti-chlamydial activity as comparedto its main active components alone, oleacein and oleocanthal, and this may be attributed to the extraction method using NaDES, a mix of natural compounds (betaine:propylene glycole), that has led to increased yield of secoiridoids” : the presence of secoiridoids is not demonstrated.
10) Line 223: remove “e” in “glycole”.
11) Line 244: a parenthesis is missing.
12) Line 255: put the numbers in Na2CO3 in subscript.
13) Line 264: replace ml by mL.
14) Some information are missing for several references. According to the author guidelines, References should be described as follows: 1. Author 1, A.B.; Author 2, C.D. Title of the article. Abbreviated Journal Name Year, Volume, page range.
15) I also notice that Rosa Sessa is cited 9 times out of the 35 references displayed, which could be assimilated to an excessively high rate of self-citations.
In conclusion, the special issue “Antibacterial Activity of Drug-Resistant Strains” deals with the antibacterial activity of drug resistant strains, but the authors do not use a resistant strain of C. trachomatis. There is therefore no match between this work and this special edition. This study should rather be improved and published in a journal that does not specifically deal with the phenomenon of resistance.
Author Response
|
Reviewer 1’s comments |
Authors’ answers |
|
In the introduction, the authors suggest that treatment failure, which can reach 23% of people treated, is due to a phenomenon of resistance. To announce this is to conceal the other reasons for this failure which are essential (re-infection, non-adherence to the treatment regimen, inadequate exposure to the antimicrobial and persistence phenomena).
|
We are sorry for the misunderstanding and we modified the relevant statements in Introduction section as following “Higher rates of treatment failures have been associated to treatment with azithromycin as compared to doxycycline towards C. trachomatis urogenital infections; drug-resistant Chlamydia serovars have been hypothesized as one of the reasons for azithromycin inefficiency, although other possible explanation may be represented by re-infection, non-adherence to therapeutic regimen, inadequate exposure to the antimicrobial drugs as well as the persistence of C. trachomatis infection [9]” (see page 2, lines 53-59).
|
|
The authors announce the use of a "green" solvent while its toxicity is demonstrated in figure S1. The choice of this solvent is then debatable. The choice of the betaine/propylene glycol mixture is not explained in the manuscript, I wonder about the possibility of replacing this mixture of solvents with a truly non-toxic solvent.
|
We thank the reviewer for the comment and for giving us the opportunity to explain. In recent years, our group has extensively studied the capacity of various NaDES to extract polyphenols from food matrices. We have developed several NaDES by combining polyols (glycerol, propylene glycol and propanediol) with aminoacids or amines (proline, betaine, choline, etc.). Propylene glycol was preferred over glycerol because it better extracts oleocanthal and other more hydrophobic polyphenols from olive oil (Di Pietro M, Filardo S, Mattioli R, Francioso A, Raponi G, Mosca L, Sessa R. Extra Virgin Olive Oil-Based Green Formulations With Promising Antimicrobial Activity Against Drug-Resistant Isolates. Front Pharmacol. doi: 10.3389/fphar.2022.885735; Francioso A, Federico R, Maggiore A, Fontana M, Boffi A, D'Erme M, Mosca L. Green Route for the Isolation and Purification of Hyrdoxytyrosol, Tyrosol, Oleacein and Oleocanthal from Extra Virgin Olive Oil. Molecules. doi: 10.3390/molecules25163654). As for the choice betaine, the decision was driven by the fact that, unlike choline, it is permitted in dermatological formulations for topical use, hence this extract could be directly used to formulate products for the topical treatment. Furthermore, upon careful analysis of the data, it may be observed that the inhibition against Chlamydia is exerted by the extracts also at dilutions far from those causing cytotoxicity in cells (1:1000; 1:10000).
|
|
Did the authors wonder about the anti-chlamydial activity of the solvent?
|
We thank the Reviewer for the remark. We did indeed study the potential anti-chlamydial activity of the NaDES alone at the same experimental conditions than the EVOO-based formulations, and we did not find any statistically significant decrease in the total number of C. trachomatis IFUs in the pre-treatment, pre-incubation and treatment phases, as compared to untreated C. trachomatis. We modified Results section as following “Initially, the anti-bacterial properties of NaDES alone against C. trachomatis were investigated in the same experimental conditions than the EVOO-based formulations; no statistically significant decrease in the total number of chlamydial IFUs was observed at either the pre-treatment, pre-incubation or treatment phases (Figure S3)” (see page 5, lines 130-133), and the Material and Methods (see pages 13-14, lines 333-379), adding these data as supplementary material (see Figure S3).
|
|
Line 86-87: the sentence seems incomplete. The word “extraction” is probably missing.
|
We added the word “extraction” to the end of the sentence as suggested by the Reviewer (see page 2, lines 96).
|
|
The authors announce a certain composition of this olive oil extract (hydroxytyrosol, tyrosol, oleacein and oleocanthal) but no experimental evidence is provided to attest to this composition. Authors must provide, supporting information, the chromatograms obtained and processed during the HPLC performed as well as the mass spectroscopy results.
|
Since these data were already presented in previously published papers, in order to shorten the manuscript, we have not included these results explicitly. We apologize for that. In the revised version of the manuscript, we have included these results adding a supplementary figure (see Figure S1). |
|
Figure 1 provides the cytotoxicity results of the olive oil extract and its 2 main components. These are diluted in the solvent to ratios where the toxicity of the latter is identified (>1:100). Under these conditions, it is therefore not possible to discriminate between the toxicity induced by the compound and that of the solvent.
|
We are sorry for the lack of clarity in the manuscript. As reported in Figure S2, the NaDES alone, at the dilution ratio of 1:100 in DMEM, showed a 21% decrease in cell viability, whereas the solution containing OOP and oleocanthal in NaDES, at the same dilution ratios in DMEM, reported much higher decrease in cell viability, namely 84.9% and 84%, suggesting that the presence of the OOPs was the main contributor to the overall cytotoxicity, with the exception of oleacein that only slightly increased the cytotoxic effect of NaDES alone (32% vs 21% decrease in cell viability, respectively). We modified the Results section as following “NaDES alone did also show a statistically significant cytotoxicity on McCoy cells up to the dilution ratio 1:200 (15.6% decrease in cell viability), albeit with a lower reduction of cell viability than that observed in the presence of OOP (58%), oleacein (16.5%) and oleocanthal (71.8%), as shown in Supplementary Figure S2” (see page 4, lines 120-123).
|
|
Table 2 provides the antibacterial activities of the compounds studied. The 1:100 column shows activity while Table 1 shows cellular toxicity of the compounds at this concentration. This activity is therefore linked to the cell death induced by the compounds and not to their antibacterial activity.
|
We are sorry for the lack of clarity in the manuscript. The antibacterial activity shown in Table 2 refers to the direct effect of EVOO-based formulations towards the extracellular C. trachomatis EBs. In particular, following 1 hour incubation at 37°C in humidified atmosphere with 5% CO2, the solution containing chlamydial EBs and the OOPs, at the dilution ratio of 1:100, was further diluted 10-times before overlaying McCoy cell monolayers for the subsequent infection, effectively reaching non-cytotoxic concentrations (1:1000). We modified the Results section as following “Then, the anti-chlamydial activity of OOP, oleacein and oleocanthal was evaluated directly against extracellular C. trachomatis EBs (pre-treatment phase), at the concentrations reported in Table 2” (see page 5, lines 134-136), and the Material and Methods as following “Subsequently, C. trachomatis EBs suspension, containing the tested compounds, was further diluted 10-times in DMEM supplemented with 10% FBS, and, hence, used to infect McCoy cell monolayers grown on glass coverslips in 24 wells cell culture trays, by centrifugation at 750 x g for 30 min at room temperature” (see page 13, lines 343-346).
|
|
The authors announce that they want to fight against the phenomenon of resistance observed in C. trachomatis. However, no experimental evidence shows that the olive oil extracts used are active against resistant strains of the bacteria. Announcing, line 211, that isolating this type of strain is time-consuming is not a plausible argument in Science.
|
We are sorry for the lack of clarity in the statement mentioned by the Reviewer, leading to confusion. Indeed, as reported in the manuscript, to date the potential chlamydial drug resistance in human infections is only suggested by the treatment failures. Unfortunately, no chlamydial resistant strains have been isolated yet, and one of the reasons may be the relatively low sensitivity (as low as 50% as compared to more modern NAATs) of culture methods for the isolation of C. trachomatis, that may be impaired by many factors, including inadequate specimen collection, storage and transport, toxic substances in clinical specimens, etc (Janssen KJH, et al. Expert Rev Mol Diagn. 2018 doi: 10.1080/14737159.2018.1498785). Therefore, this may lead to many clinical strains of C. trachomatis being unrecoverable. We modified the Discussion section as following “Consequently, the phenomenon of C. trachomatis antibiotic resistance is highly likely, although, to date, no chlamydial resistant strains have been isolated; this may be explained, indeed, by the relatively low sensitivity of culture methods for the isolation of C. trachomatis (up to 50%), especially when compared to modern nucleic acid amplification tests [31,32]” (see page 10, lines 228-232).
|
|
Line 220-222: “Interestingly, in our study, OOP showed higher anti-chlamydial activity as compared to its main active components alone, oleacein and oleocanthal, and this may be attributed to the extraction method using NaDES, a mix of natural compounds (betaine:propylene glycole), that has led to increased yield of secoiridoids”: the presence of secoiridoids is not demonstrated.
|
We apologize to the reviewer if we were not clear in drafting the manuscript. Secoiridoids such as oleuropein and ligstroside and their respective aglyconic forms (3,4-DHPEA-EA and HPEA-EA), are present in olive tree but the dialdehydic derivatives (oleacein i.e. 3,4-DHPEA-EDA, and oleocanthal, i.e. HPEA-EDA) can be found only in olive oil, since they are generated only and exclusively during the milling process. Our chromatographic analyses demonstrate that our extract contains high amounts of secoiridoids such as oleocanthal and oleacein. As already mentioned, we added these data in Figure S1.
|
|
Line 223: remove “e” in “glycole”.
|
We thank the Reviewer for the suggestion, and we removed the “e” in “glycole” (see page 10, line 242).
|
|
Line 244: a parenthesis is missing.
|
We thank the Reviewer for the suggestion, and we added the missing parenthesis (see page 12, line 265).
|
|
Line 255: put the numbers in Na2CO3 in subscript.
|
We thank the Reviewer for the suggestion, and we put numbers in Na2CO3 in subscript (see page 12, line 276).
|
|
Line 264: replace ml by mL.
|
We thank the Reviewer for the suggestion, and we replaced ml by mL (see page 12, line 285).
|
|
Some information are missing for several references. According to the author guidelines, References should be described as follows: 1. Author 1, A.B.; Author 2, C.D. Title of the article. Abbreviated Journal Name Year, Volume, page range.
|
As per Authors’ guidelines of the Journal IJMS, we did use the Mendeley Reference Manager tool for adding the references to the manuscript with the appropriate style. This tool recovers all the bibliographic information from databases like PubMed, Scopus and Web of Science, via identifiers like DOI or PMID, hence the potentially missing information from some references are more likely not available (e.g. some references are published as online only and do not have volume number and page range, like ‘Di Pietro, M.; Filardo, S.; Mattioli, R.; Francioso, A.; Raponi, G.; Mosca, L.; Sessa, R. Extra Virgin Olive Oil-Based Green Formulations With Promising Antimicrobial Activity Against Drug-Resistant Isolates. Front Pharmacol 2022, 13, doi:10.3389/fphar.2022.885735’).
|
|
I also notice that Rosa Sessa is cited 9 times out of the 35 references displayed, which could be assimilated to an excessively high rate of self-citations.
|
We are a research team devoted to the study of chlamydial organisms and their interaction with the host for the last 3 decades, hence the relatively high number of our works on this topic.
|
|
In conclusion, the special issue “Antibacterial Activity of Drug-Resistant Strains” deals with the antibacterial activity of drug resistant strains, but the authors do not use a resistant strain of C. trachomatis. There is therefore no match between this work and this special edition. This study should rather be improved and published in a journal that does not specifically deal with the phenomenon of resistance.
|
In the future, it will be necessary to assay our EVOO-based formulations against drug-resistant chlamydial isolates. However, as previously explained, no drug-resistant C. trachomatis strains have been isolated so far, although the treatment failures suggest the presence of drug-resistance in C. trachomatis. In this scenario, it is of the utmost importance to identify potential alternative therapeutic agents that may become valuable in the fight against C. trachomatis. We modified the Discussion section adding the following statements “Limitations of this study include the lack of synergism tests with the antibiotics recommended for treating chlamydial genital infections, like azithromycin or doxycycline, as well of drug-resistant strains of C. trachomatis” (see page 11, lines 255-257). |

Reviewer 2 Report
Please below for my comments:
[Abstract]
1. Please keep in mind that abstract be comprised of background, research objective, methods, results, and conclusions, and they have to be presented in a distinctive manner. Indeed authors have provided all those elements in the abstract, but there are still rooms for improvement. Authors should separate the objective and methods; the present version is confusing because the objective and methods are mixed together.
2. Avoid suggestive phrases such as “compelling evidence”. When presenting the results in the abstract, authors should be straightforward with the data. “Monotone” style is much preferred here. Moreover, please provide more numbers or quantitative data in the result part of the abstract.
3. “In conclusion, OOP may represent a promising alternative” Not sure if ‘promising’ is the correct word here. Please find other alternative.
[Introduction]
4. Scientific name of species should be presented as Italic. Please see Line 60 for “Olea europaea L.” – it has not been in Italic. Further, in the same paragraph, please add more evidence of the use of plant-based oil as antimicrobes (https://narrax.org/main/article/view/77)
5. In line 72. Please add: plant-based bioactive compounds have been previously reported to perform synergism with antibiotics in suppressing the growth of multi-drug resistant bacteria by disrupting function and structure of the membrane phospholipid bilayer. (DOI: 10.4103/japtr.japtr_111_22).
6. I get the idea that this present study is the continuation of previous project. However, novelty of the study should be EXPLICITLY stated in the introduction.
[Results]
7. Line 86—87 “which is a cultivar with 86 a high quantity of polyphenols”. Is this the finding from this present study? If this is from the previous study, please add citations.
8. Could please provide the chromatogram or MS spectra data? At least as supplementary item.
9. “Quantitative analysis of the chromatographic peaks” Did you use reference/standard compound? What are the quantitative method? You might have presented it in the method, but since it appears after the results and discussion it becomes a bit confusing. Please give a brief highlight to make it easy for readers to understand.
10. The p-value can be presented as <0.001 – it is acceptable in international writing standards.
11. Line 107 “cell viability,” please tell what cell did you use for cytotoxicity test.
12. Fig 1. Y-axis should be: Cell viability (%)
[Discussion]
13. Line 200: “green” How authors decide if the product is green or not? Is this based on subjective judgement? If so, please remove the word green.
14. Did authors perform synergism test with antibiotic, such as with azithromycin? If not, please state in the limitation.
The writing is readable, but some words/phrases need to be revised for better clarity. Authors should perform another proofreading with their peers.
Author Response
|
Reviewer 2’s comments |
Authors’ answers |
|
Please keep in mind that abstract be comprised of background, research objective, methods, results, and conclusions, and they have to be presented in a distinctive manner. Indeed authors have provided all those elements in the abstract, but there are still rooms for improvement. Authors should separate the objective and methods; the present version is confusing because the objective and methods are mixed together.
|
We thank the Reviewer for the suggestion, and we modified the abstract as following “Herein, we investigated, for the first time, the antibacterial activity against C. trachomatis of a polyphenolic extract of Extra Virgin Olive Oil (EVOO), alongside purified oleocanthal and oleacein, two of its main components, in Natural Deep Eutectic Solvent (NaDES), a biocompatible solvent. The anti-chlamydial activity of Olive Oil Polyphenols (OOPs) was tested in the different phases of chlamydial developmental cycle by using an in vitro infection model. Transmission and scanning electron microscopy analysis were performed for investigating potential alterations of adhesion and invasion, as well as morphology, of chlamydial elementary bodies (EBs) to host cells” (see page 1, lines 17-23).
|
|
Avoid suggestive phrases such as “compelling evidence”. When presenting the results in the abstract, authors should be straightforward with the data. “Monotone” style is much preferred here. Moreover, please provide more numbers or quantitative data in the result part of the abstract.
|
We agree with the Reviewer, and we modified the abstract as following “The main result of our study is the anti-bacterial activity of OOPs towards C. trachomatis EBs down to a total polyphenol concentration of 1.7 μg/mL, as showed by a statistically significant decrease (93.53%) of the total number of chlamydial inclusion forming units (p<0.0001)” (see page 1, lines 23-26).
|
|
“In conclusion, OOP may represent a promising alternative” Not sure if ‘promising’ is the correct word here. Please find other alternative.
|
We thank the Reviewer for the suggestion, and we replaced the word “promising” with “interesting” (see page 1, lines 29).
|
|
Scientific name of species should be presented as Italic. Please see Line 60 for “Olea europaea L.” – it has not been in Italic. Further, in the same paragraph, please add more evidence of the use of plant-based oil as antimicrobes (https://narrax.org/main/article/view/77)
|
We thank the Reviewer for the suggestion, and we changed all “Olea europaea L.” names in italic. We added more evidence for the use of plant-based oil as antimicrobial agents (see page 2, line 72). |
|
In line 72. Please add: plant-based bioactive compounds have been previously reported to perform synergism with antibiotics in suppressing the growth of multi-drug resistant bacteria by disrupting function and structure of the membrane phospholipid bilayer. (DOI: 10.4103/japtr.japtr_111_22).
|
We added the statement and the related reference as suggested by the Reviewer as following “Traditionally, health beneficial effects have been largely attributed to the high concentration of monounsaturated fatty acids (98–99% of the total weight of EVOO), although, in recent years, other components have been investigated, particularly olive oil polyphenols (OOP) with promising antimicrobial properties [17–20]. Moreover, plant-based bioactive compounds have been previously reported to perform synergism with antibiotics in suppressing the growth of multi-drug resistant bacteria by disrupting function and structure of the membrane phospholipid bilayer [21]” (see page 2, lines 69-75).
|
|
I get the idea that this present study is the continuation of previous project. However, novelty of the study should be EXPLICITLY stated in the introduction.
|
We agree with the Reviewer, and we modified the Introduction section as following “Herein, we investigated, for the first time, the anti-bacterial activity of OOP extracted by NaDES, as well as that of purified oleocanthal and oleacein, dissolved in NaDES, toward C. trachomatis, an obligate intracellular bacterium” (see page 2, lines 86-88).
|
|
Line 86—87 “which is a cultivar with a high quantity of polyphenols”. Is this the finding from this present study? If this is from the previous study, please add citations.
|
We apologize for not being clear enough. It is well known that Coratina cultivar is particularly rich in polyphenols, as shown by evidence in the literature. Therefore, we added the following reference to the manuscript: Mallamaci R, Budriesi R, Clodoveo ML, Biotti G, Micucci M, Ragusa A, Curci F, Muraglia M, Corbo F, Franchini C. Olive Tree in Circular Economy as a Source of Secondary Metabolites Active for Human and Animal Health Beyond Oxidative Stress and Inflammation. Molecules. 2021 Feb 18;26(4):1072. doi: 10.3390/molecules26041072. PMID: 33670606; PMCID: PMC7922482 (see page 2, line 96).
|
|
Could please provide the chromatogram or MS spectra data? At least as supplementary item.
|
As suggested by the Reviewer, we included these results as a supplementary figure (see Figure S1).
|
|
“Quantitative analysis of the chromatographic peaks” Did you use reference/standard compound? What are the quantitative method? You might have presented it in the method, but since it appears after the results and discussion it becomes a bit confusing. Please give a brief highlight to make it easy for readers to understand.
|
We thank the reviewer for bringing this to our attention. Actually, the “materials and methods” section is quite detailed, and the chromatographic method used for compounds identification and quantification is extensively described. We agree with the reviewer about the confusion that can be generated by the fact that the materials and methods section is postponed to the end of the manuscript, however we believe that, having inserted the supplementary figure S1 in this point of the manuscript, which details the method of analysis very well, we can avoid adding further detailed explanation in the text, which could make the reading heavier.
|
|
The p-value can be presented as <0.001 – it is acceptable in international writing standards.
|
We thank the Reviewer for the suggestion, and we revised the manuscript for the p-values (see page 4, lines 114-115).
|
|
Line 107 “cell viability,” please tell what cell did you use for cytotoxicity test.
|
We agree with the Reviewer, and we revised the manuscript accordingly (see page 4, lines 118 and 119).
|
|
Fig 1. Y-axis should be: Cell viability (%)
|
We apologize for the mistake, and we fixed Figure 1 Y-axis accordingly (see page 4, Figure 1).
|
|
Line 200: “green” How authors decide if the product is green or not? Is this based on subjective judgement? If so, please remove the word green.
|
The term “green solvents” mean solvents, different from the classic organic solvents such as methanol, ethyl acetate, hexane, propanol, benzene etc., with which polyphenols or other bioactive molecules are normally extracted from vegetable and food matrices. In the literature there are several papers which describe the NaDES as biocompatible solvents and belonging to the “new green chemistry”. We report, as an example, this review: Cannavacciuolo C, Pagliari S, Frigerio J, Giustra CM, Labra M, Campone L. Natural Deep Eutectic Solvents (NADESs) Combined with Sustainable Extraction Techniques: A Review of the Green Chemistry Approach in Food Analysis. Foods. 2022 Dec 22;12(1):56. doi: 10.3390/foods12010056. PMID: 36613272; PMCID: PMC9818194.
|
|
Did authors perform synergism test with antibiotic, such as with azithromycin? If not, please state in the limitation.
|
We thank the Reviewer for the suggestion. We did not perform synergism tests with antibiotic, like azithromycin, although it will be indeed very interesting to assay the potential interactions between EVOO-based formulations and the routine anti-chlamydial drugs. We modified the Discussion section adding this aspect as limitations of the study “Limitations of this study include the lack of synergism tests with the antibiotics recommended for treating chlamydial genital infections, like azithromycin or doxycycline, as well of drug-resistant strains of C. trachomatis” (see page 11, lines 255-257).
|
|
The writing is readable, but some words/phrases need to be revised for better clarity. Authors should perform another proofreading with their peers.
|
We revised the entire manuscript to improve its clarity. |

Round 2
Reviewer 1 Report
The authors answered my questions. I maintain all the same my last comment about the choice of the newspaper.